# FARS: FSM-Augmentation to Make LLMs Hallucinate the Right APIs

## Abstract

Large Language Models (LLMs) have shown remarkable ability to converse with humans and solve a wide range of tasks. They have also been extended to make use of external tools or services through API calls. This is commonly achieved by fine-tuning the model, or with the use of in-context learning, where instructions and descriptions of those external APIs, along with examples of how to call them, are given to the LLM via its prompt. Given the limited context available in the LLM prompt and other latency constraints, scaling up to a large number of tools is challenging and requires the help of an external shortlisting process to prepare instructions and examples from a large number of APIs to a smaller set of relevant ones. In this work, we propose a new way for an LLM to generate the right API calls without the need to shortlist instructions or examples. Rather, we do this by allowing the LLM to hallucinate meaningful output while grounding the generation to an available set of APIs using a finite state machine-based constrained decoding algorithm. We call our approach FARS (**FSM-A**ugmentation to make LLMs hallucinate the **R**ight API**S**). FARS allows us to ground LLMs to a large set of APIs with semantically meaningful names without using an external retriever or exemplars. We also demonstrate that with FARS, LLMs can seamlessly switch between conversation and API calling during multi-turn dialogs. We show that this can be achieved without any additional fine-tuning over the standard instruction tuning typically performed to train LLMs. This allows us to pave the way to build a truly powerful AI assistant using LLMs. We demonstrate the effectiveness of FARS for API calling on two public task-oriented API datasets: SNIPS and MultiWOZ, and a very challenging in-house Smart Home Control dataset.

## 1 Introduction

Large Language Models (LLMs) have shown remarkable ability in conversing with humans and performing a range of complex tasks. More recently, there has been an increasing trend towards adapting them to call various tools or APIs to further augment their abilities (Schick et al., 2023; Patil et al., 2023; Qin et al., 2023; Shen et al., 2023). On the commercial front, ChatGPT Plugins [1] and Bard Extensions [2] are examples of this tool augmentation, allowing an LLM to answer current affairs questions, and be able to make API calls to handle user requests. Amazon recently demonstrated a version of its Alexa voice assistant, powered by an LLM but also capable of grounding itself to real-time API calls to fetch weather information and control smart home functions [3].

To allow LLMs to work with APIs, in-context learning is used to show the LLM how to make API calls via descriptions of the API calls and demonstrations of how to use them in response to user requests. However, this method is less applicable to real world applications where the number of available APIs is too large to present as instructions or demonstrations to the LLM in its prompt. To overcome this challenge, a prompt with instructions and examples is dynamically constructed through an external shortlisting process based on the current context (Patil et al., 2023).

In this work, we propose a new way to perform the aforementioned task without the need for an external shortlisting process of API information. Rather, we take advantage of the LLM's ability to

---

[1] https://openai.com/blog/chatgpt-plugins
[2] https://bard.google.com/extensions
[3] https://www.aboutamazon.com/news/devices/amazon-alexa-generative-ai

hallucinate meaningful output while grounding this hallucination in a Finite State Machine (FSM) describing valid API calls.

Hallucination is a well-documented problem with LLMs. Despite fine-tuning and grounding with examples, they have been known to generate incorrect but logically plausible facts and completions. In the case of API calls, this results in LLMs making up API calls with arguments that sound plausible yet don't match any from the actual API catalog. We take advantage of this ability and ground the generation of the LLM to a catalog of available APIs using a finite state machine-based constrained decoding algorithm. The finite state machine consists of various states in the process of API generation and a full trace in this machine from the start state to finish state represents a valid API call. We thus effectively use the LLM to reason over state transitions and generate a trace from the FSM that corresponds to a valid API call.

We call our approach **F**SM-**A**ugmentation to make LLMs hallucinate the **R**ight API**S** or FARS in short. FARS allows us to ground LLMs to a large set of APIs with semantically meaningful names without using an external retriever or exemplars. Figure 1 shows a simplified scenario in which FARS works to generate an API call from the two available APIs - Weather and Volume, to handle a user request about weather. As typically used with LLMs, the user request is augmented with a prompt to instruct the model generate API calls along with some demonstrations. However, this prompt is static and is not a result of a retrieval process. The whole grounding process happens via constraining the generation of the LLM via an FSM containing API definitions.

A key aspect of API calls in addition to the name, arguments, and structure, is the actual argument values, which can be fixed, such as *volumeLevel* in a device control API, or mostly free text, such as *location* in a weather API. In the figure, the LLM generation is demonstrated with fixed value arguments but we also present a modification to our FSM to enable it to visit intermediate free text states and therefore generate free text arguments within the API call structure. This dual ability of being able to generate free text arguments while also having the ability to sometimes constrain the argument values is another advantage of FARS. This modification is also used to allow the chat-enabled LLM to respond to users directly with free text in addition to making API calls. Full details of our FSM approach and modifications are in Section 2.

We demonstrate the ability of FARS to generate the right API calls on two popular task-oriented API datasets - SNIPS (Coucke et al., 2018) and MultiWOZ (Budzianowski et al., 2018), and a challenging in-house Smart Home Control dataset. While SNIPS is a single-turn dataset, MultiWOZ is a complex multi-turn dataset that requires the model to reason across multiple turns in a conversation. We show that with FARS, we can achieve 93% Intent-Accuracy on SNIPS with a simple prompt without any carefully selected exemplars. An unconstrained model, with the same prompt, achieves 3%. On MultiWOZ, FARS achieves 52% Exact Match accuracy, matching fine-tuned SOTA models, while an unconstrained model gets only 26%. FARS can also be used in conjunction with retrieval and exemplars, where it resembles a few-shot constrained decoding method, but with API-specific nuances in the decoding algorithm. In this setting, FARS achieves 97% on SNIPS, compared to an unconstrained LLM that achieves 91%. We further demonstrate how FARS enables an LLM to generate both free text responses and API calls on the MultiWOZ test set. With our in-house Smart Home Control dataset, we demonstrate the API generation ability of FARS and also its ability to reason and ground itself to a large list of available devices. We achieve 46% improvement on slot accuracy and 16% on intent accuracy over unconstrained LLM with in-context instructions.

## 2 FARS: FSM-AUGMENTATION TO MAKE LLMS HALLUCINATE THE RIGHT APIS

Instruct-tuned Large Language Models have been shown to follow user requests quite well with the right information specified in the input prompt (Brown et al., 2020; Ouyang et al., 2022). If they are instructed to generate APIs, they do a good job making up a reasonable API call. They get even better when they are specified with a few relevant exemplars showing the exact APIs that are available. In both cases however, there is no guarantee that the model will generate a valid API call from the catalog of available APIs. In the exemplar case, the model will most likely do a good job adapting the seen API call, quite possible a valid one, but this very much depends on the retriever doing a good job fetching the calls.

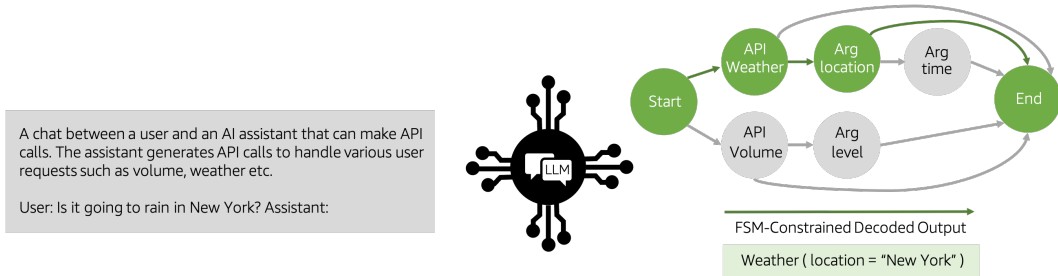

Figure 1: Grounding the LLM to generate a valid API using FARS. Generation is constrained by the simplified finite state machine shown on the right. In this instance of the user requesting information about the weather, the LLM generation is traced through the green states and transitions and the resultant text with the correct API call is at the bottom.

In this section, we first describe how an LLM typically behaves for API generation and then describe our finite state machine-augmented approach to make it generate the right APIs. We use the publicly available Vicuna-33B model (Zheng et al., 2023) as the base instruct-tuned LLM for the examples specified in this section. Although we observed similar results with our in-house instruction-tuned LLM, we share results with Vicuna-33B for reproducibility.

## 2.1 LLM HALLUCINATIONS ARE NOT ALL THAT BAD

As mentioned earlier, despite the LLM hallucinating API calls and arguments, these hallucinations make logical sense. For example, take the following simplified scenario, where the API catalog consists of the following APIs.

```
- InfoQuery to query for information about current events or specific knowledge.
  Arguments: question.
- Weather for getting weather information. Arguments: field, location.
- Search.Local for searching for places in an area.
  Arguments: placeName, location, time
  (assume these have an allowed set of values such as hospital, grocery, post office etc.)
- Volume to set volume. Arguments: level (1-10)
- Exit to leave the conversation. Arguments: none
```

Now, with the following system prompt,

```
A chat between a user and an artificial intelligence assistant that can make API calls.
The assistant generates API calls to handle various user requests such as volume, weather,
local search, info query etc.

E.g. 1:
USER: who sang blank space
ASSISTANT: InfoQuery(question="who sang blank space")

E.g. 2:
```

we see that Vicuna-33B generates the following API calls to user requests.

```
USER: find me a med emergency room in new york for tomorrow
ASSISTANT: LocalSearch(query="medical emergency room in New York for tomorrow")

USER: set volume to 50%
ASSISTANT: Volume(action="set volume to 50%")

USER: is it going to rain in new york
ASSISTANT: Weather(location="New York", queryType="forecast")

USER: okay quit
ASSISTANT: Goodbye

USER: what is the population of boston
ASSISTANT: InfoQuery(question="what is the population of boston")
```

These calls look reasonable but do not match the API catalog. Volume and Weather have incorrect arguments, the LocalSearch API call is wrong, and the model doesn't make the exit API call. This led us to believe that if we could constrain the generation to guide the model to only generate valid

API calls and arguments, we should be able to generate the right API calls. With simple trie-based constraining with preset argument values, the same LLM generates the following API calls for the same user requests with the same prompt. These API calls now exactly match the catalog.

```
USER: find me a med emergency room in new york for tomorrow
ASSISTANT: Search.Local(placeName="hospital", location="New York", time="tomorrow")

USER: set volume to 50%
ASSISTANT: Volume(level=5)

USER: is it going to rain in new york
ASSISTANT: Weather(field="rain", location="New York")

USER: okay quit
ASSISTANT: Exit()

USER: what is the population of boston
ASSISTANT: InfoQuery(question="what is the population of boston")
```

## 2.2 CONSTRAINED GENERATION WITH A FINITE STATE MACHINE

To fully extend this constrained decoding approach to real world APIs with dynamic selection of arguments, and free-text argument values, we turn to finite state machines.

A finite state machine consists of a set of finite states, and transitions between pairs of states, triggered by specific inputs. We model API generation as a finite state machine as follows.

1. We have a begin state $B$, and end state $E$. From the begin state, we are only allowed to generate an available API.

2. There are $m$ $API$ states corresponding to each of the $m$ available API calls. We transition to a particular $API$ state when the model generates that particular API call to start.

3. For each API call $API_i$ ($0 \leq i < m$) with $n_i$ possible arguments, there are an additional at $\sum_{k=1}^{n_i} {}^{n_i}P_k = \mathcal{O}(e^n)$ $ARG$ states corresponding to each permutation of possible arguments for that call. Each of these states corresponds to the state of that particular API call with a certain number of arguments filled out in a certain order, making the state space the partial permutation set of all arguments. For example, for the Weather API call from the previous scenario with arguments *field* and *location*, there are four new states - one corresponding to generating *field* as the first argument, one for *location* as the first argument, one for generating *field* first and *location* second, and one for generating *location* first and *field* second. We chose this design instead of having just a single fixed order of all arguments in the API call since we experimentally found that the LLM does a better job when we let it choose the order and subset of arguments to predict.

4. To allow the model to skip arguments, we allow transitions from every $ARG$ state to the end state $E$. This way, the API call will consist only of arguments generated until that state.

5. Argument values can either be chosen from a set of predetermined values such as levels for the Volume API, or can be completely free-text such as the question for an InfoQuery call. We therefore have, for each $ARG$ state, an additional value state - either $V^{fix}$ for generating an argument value from a fixed set, or $V^{un}$ to allow unconstrained generation. From each $ARG$ state, we transition to the $V$ state when we start generating the value. Once generation is complete, we can transition to the next $ARG$ state or the end state $E$.

6. To allow generation of multiple concurrent API calls, e.g., switching off the light and reducing air temperature, we introduce final state $F$. After the API end state $E$, we can either transition to the begin state $B$ to generate another API call or transition to this final end state $F$ to signal the end of the whole API generation process.

In total, given $m$ APIs, each with $n_i$ arguments, the complete set of states $S$ is given by

$$S = \{B, E, F\} \cup \{API_i : 0 \leq i < m\} \cup \{ARG_{is} : 0 \leq i < m, s \in \psi(\{0, 1, \ldots n_i\})\}$$
$$\cup \{V_{is}^{un|fix} : 0 \leq i < m, s \in \psi(\{0, 1, \ldots n_i\})\}$$

where $\psi(X)$ is the set of all partial permutations of elements in set $X$.

Figure 2 shows a graphical representation of the FSM we just described. The begin, end, and final states are marked in green. The $API$ states are marked in blue. We expanded two of them

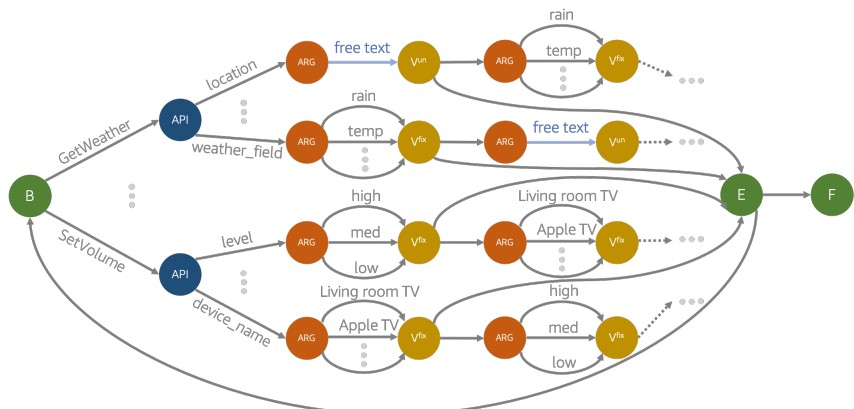

Figure 2: An example Finite State Machine for API generation. States corresponding to two APIs and two of their arguments are shown expanded while all the other states are collapsed with dots.

- *GetWeather* and *SetVolume* here. All the $ARG$ states are marked in orange. As shown, we can transition to any of the allowed argument states from the $API$ state first and from there, we can then transition to one of the remaining argument states. The argument value states are all marked in yellow as $V$. Unconstrained value states allow generating free text (e.g. *location*) and the ones with a fixed set of values (e.g. *level*) only allow generation of those values. When we attach this finite state machine to the LLM while decoding, the generation is constrained to only allow feasible tokens that make up valid API calls due to the fixed states and state transitions.

While the FSM enforces structure, it doesn't perform any reasoning on which of the available API calls or arguments to generate. That is completely left up to the parametric knowledge of the LLM. In order to describe the some of the other logical transitions of the FSM, such as how to end free text argument value generation, we need to describe that to the LLM through instructions in the prompt. We describe further implementation details in Section 3.1

**Integrating with a Conversational Experience** The FSM just described, once engaged, always forces the LMM to predict an API call from the catalog. However, in an AI assistant application, the LLM would also generate responses to the user in addition to making API calls. To facilitate this, we augment the FSM with a non-API state which allows unconstrained free-text generation. We can choose to allow the LLM to follow the non-API route or API-grounding route. With this modification, we enable the LLM to both respond directly or generate a valid API when necessary.

## 3 EXPERIMENTAL DETAILS

### 3.1 IMPLEMENTING AND OPTIMIZING FARS

In this section we describe how to efficiently implement FARS using a constrained decoding algorithm that relies on a trie structure, dynamically built to follow the FSM defined earlier.

We first store all APIs, corresponding arguments, their types (categorical/free-text), and possible values in an API bank. From this, we construct our trie. A trie is a type of *k-ary* search tree that is used to efficiently store all possible sequences in a set. It can be used to query the possible next items given a certain prefix, akin to a compact hash-table. Our trie is defined at the token level to work with LLM generation, and it determines the allowed next tokens that the model can generate based on the prefix of tokens generated so far, following the constraints of the FSM, defined by the schema in the API bank. Built dynamically, the trie contains *holes* that correspond to the unconstrained state in the FSM where we allow free text generation to generate free-text argument values. Each hole is defined by a special hole-start and hole-end token in the trie, conditioned on the argument it is generating the value for. When the LLM generates the hole-start token for that particular argument, we relax the constraints and let the model generate free text until it generates the hole-end token.

We use a dynamic trie since a simple trie with a predetermined set of all possible sequences is prohibitively large due to a) the exponential combinations of arguments - $\mathcal{O}(e^n)$, and b) the infinite possibilities while generating multiple API calls in the same turn. With the dynamic trie, we overcome these issues by building it dynamically for only a few steps until the next argument or the next API call is ready to be generated. When the LLM is ready to generate further, we extend the trie to include the possible next set of arguments or API calls to choose from. This reduces the overall size of the trie to a polynomial $\mathcal{O}(n^2)$. Algorithm 1 describes the pseudocode of our approach. In our implementation, we initialize the trie to cover all API call names. During inference, other arguments and possible API calls are chosen from the API bank and added on dynamically. For any changes to the API schema such as adding or modifying API names, arguments, or values, we simply need to update the API bank and re-initialize the trie with the new set of API names.

---

**Algorithm 1** Constrained API Generation with a Dynamic Trie with Holes

---

**Input:** Trie $T$, LLM $M$, Generated Sequence $S = [\,]$, Unconstrained start and end tokens $U_s, U_e$
  **def** $Next(M, S, T^*)$ - Returns next token generated by $M$, constrained by $T$, with prefix $S$.
  **def** $Complete(S, T)$ - Returns $True$ if $S$ is complete w.r.t sequences in $T$.
  **def** $GenerateAnotherARG(S)$ - Returns $True$ if $S$ is ready to generate another argument.
  **def** $GenerateAnotherAPI(S)$ - Returns $True$ if $S$ is ready to generate another API.
**Steps:**
  1: Initialize $T$ with all possible API names
  2: $constrained = True$
  3: **while not** $Complete(S, T)$ **do**
  4:    **if** $constrained$ **then**
  5:        $c_t = next(M, S, T)$
  6:    **else**
  7:        $c_t = next(M, S)$
  8:    **end if**
  9:    $S = S + [c_t]$
 10:    **if** $c_t == U_s$ **then**
 11:        $constrained = False$
 12:    **else if** $c_t == U_e$ **then**
 13:        $constrained = True$
 14:    **end if**
 15:    **if** $GenerateAnotherARG(S)$ **then**
 16:        Extend $T$ by one step to include each of the remaining args
 17:    **end if**
 18:    **if** $GenerateAnotherAPI(S)$ **then**
 19:        Extend $T$ by one step to include all possible API names
 20:    **end if**
 21: **end while**

---

## 3.2 BASE LLM, SETTINGS, AND EVALUATION DATASETS

Although we obtain similar results with an in-house LLM, we present results using Vicuna-33B since it is publicly available and helps in reproducing our results. We keep the wording of the prompt that the Vicuna model is fine-tuned with, but modify it to include information about API generation. Our base prompt for API evaluation for all the datasets is the following. Information about the API calls and examples (indicated by xxx) is filled out specifically for each dataset.

```
A chat between a user and an artificial intelligence assistant that can make API calls.
The assistant generates API calls to handle various user requests such as xxx.

E.g 1:
USER: xxx
ASSISTANT: xxx

E.g 2:
```

We evaluate FARS on three datasets - SNIPS (Coucke et al., 2018), MultiWOZ (Budzianowski et al., 2018), and an in-house Smart Home Control Dataset. SNIPS consists of single-turn user utterances and target API calls across seven different intents such as weather, music, restaurant etc. MultiWOZ

| | IA | SR | SF1 | IA | SR | SF1 | IA | SR | SF1 | IA | SR | SF1 |
|---|---|---|---|---|---|---|---|---|---|---|---|---|
| | **Playlist** | | | **Restaurant** | | | **Weather** | | | **Music** | | |
| | No API retrieval | | | | | | | | | | | |
| Unconstrained LLM | 5.0 | 10.0 | 65.8 | 0.0 | 1.0 | 14.9 | 8.0 | 8.0 | 41.7 | 6.0 | 11.0 | 41.7 |
| FARS (Ours) | 100.0 | 56.0 | 72.8 | 90.0 | 35.0 | 61.4 | 99.0 | 43.0 | 66.9 | 94.0 | 43.0 | 50.4 |
| | With API retrieval | | | | | | | | | | | |
| Unconstrained LLM | 91.0 | 59.0 | 70.3 | 91.0 | 37.0 | 63.7 | 95.0 | 45.0 | 70.3 | 87.0 | 38.0 | 48.6 |
| FARS (Ours) | 100.0 | 77.0 | 82.7 | 97.0 | 47.0 | 68.0 | 100.0 | 55.0 | 72.2 | 99.0 | 51.0 | 63.7 |
| | **Book** | | | **Creative** | | | **Screening** | | | **Overall** | | |
| | No API retrieval | | | | | | | | | | | |
| Unconstrained LLM | 2.0 | 0.0 | 0.0 | 0.0 | 0.0 | 0.0 | 0.0 | 7.0 | 25.2 | 3.0 | 5.3 | 27.0 |
| FARS (Ours) | 100.0 | 55.0 | 75.1 | 94.0 | 75.0 | 77.1 | 78.0 | 37.0 | 53.4 | 93.6 | 49.1 | 65.3 |
| | With API retrieval | | | | | | | | | | | |
| Unconstrained LLM | 98.0 | 56.0 | 75.8 | 82.0 | 71.0 | 72.6 | 96.0 | 54.0 | 49.7 | 91.4 | 51.4 | 64.4 |
| FARS (Ours) | 100.0 | 55.0 | 80.2 | 86.0 | 74.0 | 73.8 | 99.0 | 60.0 | 69.4 | 97.3 | 59.9 | 72.9 |

Table 1: Intent Accuracy (IA), Slot Recall (SR), and Slot F1 (SF1) metrics on SNIPS comparing FARS and an unconstrained LLM, with and without API exemplar retrieval.

is a more complex dataset that consists of multi-turn user conversations with API calls and also assistant responses after API call execution. We primarily report numbers on API call performance but also show some examples of how we can disengage our FSM and allow the model to generate responses freely. Finally, we evaluate FARS on a challenging in-house Smart Home control dataset consisting of around 60 different API calls and a large number of devices as arguments.

# 4 RESULTS

## 4.1 MAKING SINGLE-TURN API CALLS - SNIPS

We evaluate our approach by employing two inference settings on the SNIPS dataset: (1) **No API retrieval** where there are no relevant exemplars or API descriptions. (2) **With API retrieval** where there are API descriptions and domain-specific exemplars. We benchmark FARS against the unconstrained model, our baseline. We report intent accuracy, slot recall, and slot F1 score on SNIPS. Slot recall measures whether all the gold slots have been predicted correctly by the model.

As seen in Table 1, without API retrieval, FARS significantly outperforms unconstrained LLM by 90.6, 43.9 and 38.3 absolute points averaged across 7 intents on intent accuracy, slot recall and slot F1 score respectively. We observe drastic improvements in performance as an unconstrained LLM without available API information in the prompt severely hallucinates incorrect intents and slots whereas our approach effectively predicts correct hypotheses as it is grounded by the FSM.

In the case where we retrieve and provide relevant API exemplars in the prompt, the unconstrained LLM improves as the model is now aware of the API calls. Our retrieval is based on sentence similarity scores between the API description and user utterance but we also make it an oracle setting where the right API call is always inserted, giving us an upper bound. FARS without API retrieval achieves similar performance to the unconstrained LLM with oracle API retrieval, emphasizing the effectiveness and efficiency of our approach. We find that a big portion of errors in FARS are from the screening domain, where it is worse than the unconstrained LLM with retrieval. Upon further error analysis, we discovered that this domain has certain arguments that weren't easy for the model to predict without explicit examples. For example, *object_type* with values such as *movie schedules* and *movie times*. When provided with API exemplars, FARS further improves upon its unconstrained counterpart by 5.9, 8.4 and 8.4 absolute points on the same metrics, showing that FARS still provides an advantage when combined with retrieval.

## 4.2 SMART HOME API CALLS WITH DEVICE CONTROL

We report intent accuracy and slot accuracy on our internal Smart Home Control dataset. Since this is a proprietary dataset, we report relative numbers. This dataset contains single turn queries for

|  | Intent | Slot |
|---|---|---|
|  | All devices | |
| Unconstrained LLM | baseline | |
| FARS (Ours) | +15.9 | +46.3 |
|  | With top-k device retrieval | |
| Unconstrained LLM | +16.4 | +9.2 |
| FARS (Ours) | +27.2 | +56.2 |

Table 2: Intent and Slot Accuracy for Smart Home Control APIs. We show relative improvement over baseline unconstrained LLM in two scenarios - with and without shortlisting top-k devices.

controlling various home devices. Each device has its respective APIs, sometimes shared among devices. Given this setup each home may have K devices and M ($>> K$) APIs. This is an especially hard dataset since there are many semantically similar APIs such as *Adjust_Setting*, *Brightness_Increase*, and different set of slots for each API e.g. *(mode, level)* v/s *(setting, setting_value)*.

Table 2 shows that by constraining LLM with FARS, we see an improvement of 46.3 points on slot accuracy and 16 points on intent accuracy. Reducing search space by shortlisting top devices and APIs further improves performance. With reduced search space, an unconstrained LLM matches performance of the full search space FARS on intent accuracy. However, FARS vastly outperforms it in slot accuracy since each API has slightly different slots names, and by constraining with an FSM, our approach is able to handle these nuances better. FARS with reduced search space is the best overall model providing a further 10 point improvement over FARS with full search space.

### 4.3 Multi-turn API Calls and Responses - MultiWOZ

On the MultiWOZ 2.2 dataset, we evaluate FARS without any special exemplar retrieval and find that we match the turn-level Full-Signature (exact match) accuracy of three SOTA models[4] with a score of 52.2 as seen in Table 3. We also report intent accuracy and slot recall of our model for reference. We achieve these scores without additional training and by simply engaging the FSM during generation at each turn in the conversation where we need to generate an API. An unconstrained LLM achieves a score of 26.3, so FARS helps us double the score.

In addition to API calling, we also enable the LLM to generate direct assistant responses when required by disengaging the FSM. Since we do not train the model and only constrain it during generation, it retains its generic instruction following and conversational ability and we see this on display during the prediction of assistant responses. We achieve an average BERTScore (Zhang et al., 2020) of 0.90. Additionally, we often found that our model responses were of higher quality than the reference. Take the following turn for example. Our model, being a powerful instruction following model with chat abilities, generates a more helpful answer than the gold response by fetching information from previous turns in the conversation.

```
...conversation about address and information on the varsity restaurant...
USER: I do not need a reservation for the restaurant just information.
GOLD ASSISTANT: Ok. Do you need anything else then?
PRED ASSISTANT: I apologize for the confusion. Here is the information you requested:
The Varsity Restaurant, 35 Saint Andrews Street City Centre, Postcode: cb23ar
Phone: 01223356060
Please let me know if there is anything else I can help you with.
```

During our error analysis for API calling, we found a fair amount of noise in the test dataset, based on which our model predictions were penalized. There were many turns with multiple API calls where the additional call wasn't warranted and there were also turns that had a logically different structure. Take the following turn for example.

```
...conversation about visiting Magdalene College on Magdalene Street and Dojo Noodle bar...
USER: No, I was just curious. I am also looking for a taxi between those two places.
GOLD ASSISTANT: Taxi.find_taxi()
PRED ASSISTANT: Taxi.find_taxi(destination="magdalene street", departure="dojo noodle bar")
```

---

[4]Scores for the SOTA models were fetched from the MultiWOZ 2.2 paper (Zang et al., 2020)

|                                    | Intent | Slot | Full-Signature |
|------------------------------------|--------|------|----------------|
| Unconstrained LLM                  | 55.9   | 31.8 | 26.3           |
| FARS (Ours)                        | 85.0   | 61.9 | 52.2           |
| TRADE (Wu et al., 2019)            | -      | -    | 45.4           |
| SGD-Baseline (Rastogi et al., 2020)| -      | -    | 42.0           |
| DS-DST (Zhang et al., 2019)        | -      | -    | 51.7           |

Table 3: Metrics on MultiWOZ conversational dataset comparing FARS to SOTA models. We match SOTA models on full-signature accuracy with no training or retrieval.

Our model was penalized here for inferring the departure and destination arguments from earlier in the conversation. In the gold conversation, these were explicitly elicited from the user in the next turn. Our model, knowing the arguments, fills them in without an additional turn.

## 5 RELATED WORK

There is an increasing amount of literature exploring API tool usage with LLMs. This includes work such as Toolformer Schick et al. (2023), Gorilla (Patil et al., 2023), ToolLLM (Qin et al., 2023), HuggingGPT (Shen et al., 2023), and TaskMatrix (Liang et al., 2023). Most of these approaches rely on including examplars and instructions in the input prompt to the LLM. This makes them dependent on a retriever or shortlister to fetch a few examples to include in the prompt. With FARS, we don't need to do this. The finite state machine contains information about the API catalog and we can simply engage it during generation and include simple instructions in the prompt.

Our constrained decoding algorithm is similar to the work on autoregressive entity retrieval by De Cao et al. (2020), in which the authors ground the generation of a trained autoregressive model to generate entity links from an existing catalog of entities. In our work, we follow a similar approach to grounding, but tackle the problem of API call generation. We also integrate this approach with instruct-tuned LLMs to completely skip any training and simply guide the model during inference.

Constrained Decoding for semantic parsing is another closely related field of work (Shin et al., 2021; Wu et al., 2021; Rongali et al., 2022). Here, the focus is primarily on using constrained decoding to enforce structure after training, or in an in-context learning setting, which requires fine-grained retrieval. The style of constraining in these works, using a simple trie, also doesn't generalize to API calling and free-text *holes* for arguments. We propose a more general framework and show how to combine it with an instruct-tuned LLM, which can then be prompted to generate the right APIs.

## 6 CONCLUSION AND FUTURE WORK

We propose FARS, an FSM-augmented approach to make LLMs generate API calls from a catalog without any retrieval. Our approach allows us to ground LLM generation to a large set of APIs with semantically meaningful names. We provide a formal framework for FARS and also describe our implementation in detail. We demonstrate the effectiveness of our approach on three different datasets - SNIPS, MultiWOZ (multi-turn), and an in-house Smart Home Control dataset, where we show that FARS achieves massive improvements over an unconstrained LLM.

An important limitation of FARS currently is that the model generation is always grounded to the available API catalog. So, if a user requests an action that isn't covered by the catalog, the model will generate the closest one from the existing APIs. This scenario can be handled better by exploring confidence scores of generated API calls and determining whether the grounding is reasonable. Further, we briefly demonstrated how to enable both free text response generation and grounded API calling using MultiWOZ but these decisions were predetermined i.e. we knew when to engage the FSM and when to not. An important next step would be to demonstrate how to allow the model to make this decision by providing some instructions in the prompt and setting up the FSM appropriately. We leave these directions to future work.

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

## A    APPENDIX: INFERENCE DETAILS

We use HuggingFace generation inference for our experiments. We create a lambda function and pass it to the *prefix_allowed_tokens_function* argument in *generate* function. The lamda function is initialized from the trie, which is in turn initialized from our API bank schema, and consists of API name sequences at the start. The lambda function contains logic that dynamically adjusts the trie and queries the latest state of the trie to return the valid set of tokens for generation at each step. We use greedy decoding with beam size 1 for API generation in all our experiments, both constrained and unconstrained, so there is no sampling or temperature set. We set the *max_new_tokens* to 200.

## B    APPENDIX: LATENCY ANALYSIS

There are some caveats to our implementation which prevent an easy, straightforward comparison between FARS and an unconstrained LLM. A quick note, since we use HuggingFace generation inference for our experiments, it may not reflect the run-time usage of models, which usually takes place on a more optimized inference engine. Coming to the caveats, first, we noticed that constrained generation ensures the model terminates right after finishing an API call with the proper set of arguments but an unconstrained model may generate a lot more words - sometimes it hallucinates additional arguments and sometimes it just does not terminate and generates new conversation turns. In such cases, we saw smaller times for constrained generation, in spite of the additional processing required for constraining. Second, the majority of our processing time is in building the dynamic trie to avoid the $n!$ explosion that comes with pre-building the entire trie. Once built however, the additional overhead is just a trie lookup, which is extremely fast and practically adds no overhead. We see this in our evaluation on a dataset where in the first few examples where the trie is being built, there is a larger overhead but later, the differences disappear.

Now, under similar conditions with similar token lengths generated and no post-API turn hallucination with unconstrained models, with HuggingFace inference, we found that there was no difference between constrained and unconstrained models. On SNIPS, the average time per sample for unconstrained was 4.27 seconds while for constrained, it was 4.39 seconds (3% higher). Table 4 provides further breakdown of these times. In a production setting, where the trie already contains most of the sequences, there would be practically no overhead from just a sequence of trie lookups and constraining could in fact be faster since the model isn't allowed to hallucinate too many additional arguments. Furthermore, despite not being implemented in our current work, there is an opportunity to skip model generation and *fast-track* the generated sequence whenever there is only one possible path in the trie. This has potential to greatly speed up generation time further.

| Intent | Unconstrained | | Constrained (FARS) | |
|---|---|---|---|---|
| | Avg. Time | SPM | Avg. Time | SPM |
| Playlist | 3.86 | 15.53 | 3.96 | 15.15 |
| Restaurant | 5.98 | 10.03 | 6.27 | 9.57 |
| Weather | 4.33 | 13.84 | 4.67 | 12.84 |
| Book | 4.28 | 14.01 | 4.38 | 13.69 |
| Creative | 2.44 | 24.55 | 2.51 | 23.94 |
| Screening | 5.08 | 11.82 | 5.08 | 11.81 |
| Music | 3.94 | 15.24 | 3.87 | 15.50 |
| Overall | 4.27 | 15.00 | 4.39 | 14.64 |

Table 4: Average Time and Samples per minute (SPM) comparing unconstrained generation and constrained FARS generation on examples from the SNIPS dataset with HuggingFace inference.

