# OpenReview forum: "FARS: FSM-Augmentation to Make LLMs Hallucinate the Right APIs"
_ICLR.cc/2024/Conference — Submitted to ICLR 2024_

### Official Review · Reviewer_NqbF · 2023-10-31

**Soundness:** 3 good
**Presentation:** 3 good
**Contribution:** 3 good
**Rating:** 8
**Confidence:** 4

**Summary:**

Generative models may hallucinate APIs when generating their responses. To this end, this paper proposes FARS to use constrained decoding to limit the token selection during API calls, making LLMs generate desired API formats. Specifically, the authors construct a finite state machine to enforce the decoding to follow the structure Begin-API-Argument-Value-End and limit the number of available tokens to the number of designed states (e.g., # APIs when generating APIs).  The implementation is based on the dynamic trie implementation to save the space cost. Experiments show the effectiveness of FARS compared with unconstrained LLMs.

**Strengths:**

1. FARS is a novel approach that involves an inference-time intervention to enforce LLMs to generate the correct API formats
2. The approach is sound and well-illustrated.
3. Experiment results show the improvement is quite apparent.

**Weaknesses:**

1. Lack of wall time analysis. I am not sure if this method will bring much extra time cost since the approach seems to involve many CPU operations. It will be good to add a wall time comparison.
2. The approach is more like eliminating "syntax" error but not "semantic" error. Will FARS eliminate "syntax" error but increase "semantic" error? For example, the function is originally semantically correct but with a wrong format, whereas FARS corrects the syntax but brings semantic errors. It will be good to see how much performance is obtained by "syntax" correction and if FARS introduces more "semantic" errors.
3. Lack of some implementation details. Please see questions for details.

**Questions:**

What temperature is used in the experiments? The comparison may be less convincing if the temperature is high for baseline models.

---

> ### Author Response · Authors · 2023-11-17
>
> ----Lack of wall time analysis----
>
> In general response
>
> ----Will FARS eliminate "syntax" error but increase "semantic" error?----
>
> In our implementation, we simply constrain the allowed token set without changing any of the probability scores. The highest probability token will remain the highest after constraining, provided it doesn’t interfere with the allowed arguments, API names, or format. Now, when we constrain and force the model to choose a token from the valid subset of tokens, which is different from the original global max, the question of whether that affects the rest of the generation is a hard one to answer in theory. We believe a good model should be able to generate a good continuation from any prefix, so we would argue that no, it doesn’t introduce any more semantic errors. To support this, we ran a quick empirical study on the MultiWOZ dataset on how many individual samples the unconstrained more performs better than the constrained model (not just the average, on which we know the constrained model is clearly better). We found that over 1000 dialogs and 7372 API turns, the unconstrained model is better than constrained just 6 times (0.08%).
>
> ----What temperature is used in the experiments? The comparison may be less convincing if the temperature is high for baseline models.----
>
> We don’t use temperature at all in our experiments. We use greedy decoding for API Generation since we don’t care about sampling diversity and just want the API call to be predicted correctly. This is for both constrained and unconstrained models. In the public baseline numbers for MultiWOZ from other previous works, we verified that it was also greedy, without any temperature sampling (those are all smaller models SFTed for this task anyway, not text generation LLMs). We included a section in the appendix about these inference details.

---

> > ### Comment · Reviewer_NqbF · 2023-11-20
> >
> > Thanks for the clarification. I raised my score to 8.

---

> > > ### Author Response · Authors · 2023-11-21
> > >
> > > Thank you for your consideration!

---

### Official Review · Reviewer_msYT · 2023-11-01

**Soundness:** 4 excellent
**Presentation:** 4 excellent
**Contribution:** 3 good
**Rating:** 6
**Confidence:** 4

**Summary:**

The paper proposes a new approach called FARS to enable Large Language Models (LLMs) to generate the right API calls without the need for shortlisting instructions or examples. The approach uses a finite state machine-based constrained decoding algorithm to ground the generation of LLMs to a set of available APIs. The paper demonstrates the effectiveness of FARS on three datasets - SNIPS, MultiWOZ, and a Smart Home Control dataset, showing significant improvements over an unconstrained LLM.

**Strengths:**

1) The paper introduces a novel approach to address the problem of generating the right API calls without shortlisting instructions or examples.
2) The use of a finite state machine-based constrained decoding algorithm provides a structured and grounded approach to API generation.
3) The experimental results on the three datasets demonstrate the effectiveness of FARS, showing significant improvements over an unconstrained LLM.

**Weaknesses:**

1) The paper could provide more details on the implementation of FARS, including the specific steps and algorithms used to integrate the finite state machine with the LLM.
2) The paper lacks a thorough discussion of the limitations and potential future directions of the proposed approach.

**Questions:**

see weaknesses

---

> ### Author Response · Authors · 2023-11-17
>
> ----The paper could provide more details on the implementation of FARS.----
>
> In general response
>
> ----The paper lacks a thorough discussion of the limitations and potential future directions of the proposed approach.----
>
> Thank you for bringing this to our notice and apologies for lack of more information on this. Due to space constraints, we limited this discussion to just the last paragraph of our paper. We however mention two concrete limitations and future directions to address them - 1) our model’s inability to handle APIs from unknown domains due to always constraining it to an API bank and 2) lack of analysis on the model’s ability to conditionally choose to engage the FSM. We will add a section on this in the appendix after further discussion among ourselves.

---

> > ### Author Response · Authors · 2023-11-21
> >
> > Just checking in since the discussion period is coming to a close. Please let us know if our responses were satisfactory and if you have any further questions/analyses you want us to run.

---

> > > ### Comment · Reviewer_msYT · 2023-11-22
> > >
> > > Thank you for the explanation of the limitations of this paper. This has resolved my confusion, and I will maintain my rating.

---

> > > > ### Author Response · Authors · 2023-11-22
> > > >
> > > > Thank you for your consideration!

---

### Official Review · Reviewer_QTFS · 2023-11-06

**Soundness:** 4 excellent
**Presentation:** 4 excellent
**Contribution:** 4 excellent
**Rating:** 6
**Confidence:** 4

**Summary:**

The paper introduces a novel approach called FSM-Augmentation to make Language Models generate correct API calls by grounding their output in a Finite State Machine that describes valid API calls. This method, FARS, aims to address the issue of LLMs "hallucinating" plausible but incorrect API calls by using a constrained decoding algorithm based on FSM.

FARS's approach allows for the dynamic selection of API arguments and free-text values, improving upon traditional methods that rely on fixed argument orders. The FSM design enables the LLM to predict the order and subset of arguments, enhancing flexibility and accuracy.

**Strengths:**

1. The paper introduces a finite state machine-augmented approach, which is an improvement over traditional LLMs that often hallucinate plausible but incorrect API calls. By grounding the LLM's generation process in a finite state machine, the model is constrained to produce only valid API calls, which is a practical solution to a common problem in LLM outputs.

2. No Need for External Retrievers: Unlike other methods that rely on external retrievers or exemplars to guide the generation of API calls, FARS operates independently by incorporating the API catalog information into the FSM. This reduces the complexity and potential points of failure associated with external dependencies.

**Weaknesses:**

1. The effectiveness of FARS is contingent on the FSM's knowledge of the available API catalog. It is not clear how easy to update FSM if there is a new API function added
2. Potential Overhead during inference. Creating and updating the FSM to reflect the current state of API offerings could introduce overheads. How fast is the inference speed of FARS compared with unconstrained LLM?
3. The scope of the paper's evaluation is limited to the Vicuna-33B model's performance on specific datasets. A broader assessment across various models would provide a more comprehensive understanding of FARS's effectiveness and generalizability.

**Questions:**

1. What is the retrieval model used in API retrieval setting?

---

> ### Author Response · Authors · 2023-11-17
>
> ----It is not clear how easy to update FSM if there is a new API function added----
>
> Updating the FSM with the new API is very easy. In fact, it is one of the advantages of our approach since we don’t have to touch the model or exemplars at all.
> We currently use a dynamic trie structure to represent the FSM. This trie is constructed from a schema dictionary that contains all the APIs, arguments, and allowed values. To add a new API, we can simply add it to the dictionary and re-run the trie initialization. This adds the sequence corresponding to generating that API name to the trie and includes the possible arguments in a tracker. Since the trie is dynamically expanded to cover arguments as the sequence keeps progressing during inference, we already have logic to go through all available APIs and extend the trie so nothing needs to be done for the new API there. We added these details in the paper to make this clear.
>
> ----How fast is the inference speed of FARS compared with unconstrained LLM?----
>
> In general response
>
> ----The scope of the paper's evaluation is limited to the Vicuna-33B model's performance on specific datasets----
>
> We performed evaluation with internal models of different sizes and found similar trends as well but reported only on Vicuna for reproducibility. Our approach required us to change the underlying generation strategy so we couldn’t go with closed models such as ChatGPT, where we don’t have access to the weights to run generation ourselves. We required an instruct-tuned LM and we believed that Vicuna-33B was a good strong candidate to go with. We also briefly experimented with LLAMA-2-Chat but found that that model was very resistant to generating APIs, no matter how we prompted it with exemplars.
>
> ----What is the retrieval model used in API retrieval setting?----
>
> We use a simple sentence-encoder similarity approach between the API calls and utterances to choose the retrieved set but also ensure that the gold exemplar (one with the target API call) is always present in SNIPS. This especially favors the unconstrained model since the exemplars are the only source of knowledge for it. We included info about this in the original draft and also updated it to include more details. For the internal smart home dataset, we didn’t perform retrieval for APIs, since each device came with a list of possible APIs or actions and we could just use those. For device retrieval/shortlisting, we used the same sentence transformer for similarity computation between the device details and user utterance and again also ensured that gold ones are present.

---

> > ### Author Response · Authors · 2023-11-21
> >
> > Just checking in since the discussion period is coming to a close. Please let us know if our responses were satisfactory and if you have any further questions/analyses you want us to run.

---

### Author Response · Authors · 2023-11-17
**General response to all reviewers**

We thank all the reviewers for their insightful comments and suggestions. We appreciate the detailed questions, some of which led us to perform and include new analyses to help improve our work. We responded to specific questions from each reviewer in-line but overall, there seem to be two major points of discussion, which we address here.

First, there was the question of inference speed/wall clock time comparison. This is a great point but there are some caveats to our implementation which prevent an easy, straightforward comparison between FARS and an unconstrained LLM. A quick note, we use HuggingFace generation inference for our experiments, which may not reflect the run-time usage of models, which usually takes place on a more optimized inference engine. Coming to the caveats, first, we noticed that constrained generation ensures the model terminates right after finishing an API call with the proper set of arguments but an unconstrained model may generate a lot more words - sometimes it hallucinates additional arguments and sometimes it just does not terminate and generates new conversation turns. In such cases, we saw smaller times for constrained generation, in spite of the additional processing required for constraining. Second, the majority of our processing time is in building the dynamic trie to avoid the n! explosion that comes with pre-building the entire trie. Once built however, the additional overhead is just a trie lookup, which is extremely fast and practically adds no overhead. We see this in our evaluation on a dataset where in the first few examples where the trie is being built, there is a larger overhead but later, the differences disappear.

Now, under similar conditions with similar token lengths generated and no post-API turn hallucination with unconstrained models, with HuggingFace inference, we found that there was no difference between constrained and unconstrained models. On SNIPS, the average time per sample for unconstrained was 4.27 seconds while for constrained, it was 4.39 seconds (3% higher). In a production setting, where the trie already contains most of the sequences, there would be practically no overhead from just a sequence of trie lookups and constraining could in fact be faster since the model isn’t allowed to hallucinate too many additional arguments. Furthermore, despite not being implemented in our current work, there is an opportunity to skip model generation and “fast-track” the generated sequence whenever there is only one possible path in the trie. This has potential to greatly speed up generation time further. We included a section with a table on the times taken and additional discussion around this in the appendix.

Second point of discussion was some lack of implementation details. We have duly noted this. We updated the draft to add more details. Summarily speaking, for this work, we use a dynamic trie to simulate the FSM. The trie is populated with token sequences representing various API call invocations. We use certain reserved token markers to represent free-text values and also to signal when to expand the trie to the next set of arguments or APIs (constructing the full trie with all permutations of argument sequences and APIs is infeasible). We originally refrained from too much instruction on this exact implementation approach since there could be other ways to implement it, depending on how your API bank is structured (you could simply pre-construct the entire trie in some cases) and what kind of kernel this is being run on.

---

### Meta-Review · Area_Chair_RohA · 2023-12-06

**Metareview:**

This paper proposes to constrain the output space of LLMs with a finite-state machine that encodes knowledge about the universe of APIs in the context of tool-using LLMs. This approach is found to outperform unconstrained LLMs by a significant margin, especially in cases where the API prediction has to be done zero-shot (i.e., without retrieval).

Given the increased importance of calling tools within LLM generations, the paper tackles an important problem. However, despite the generally favorable reviews, after reading the paper I think there are several weakness that prevent me from recommending acceptance. In particular:
- The method is only tested on a single LLM.
- The method is largely tested against internal baselines. The only benchmark in which it compares against previous work is on MultiWOZ (Table 3). But they leave out works that perform much better (e.g., see https://paperswithcode.com/sota/multi-domain-dialogue-state-tracking-on-1)
- There has been much work on performing constrained decoding with LLMs (as cited by the authors in the related work section), but they describe these works by saying that "The style of constraining in these works, using a simple trie, also doesn’t generalize to API calling and free-text holes for arguments." I don't think this is accurate. Some constrained decoding works decode with constraints from context-free grammars (e.g., Shin et al. 2021), so in some sense this work (which only uses FSM constraints) is a special case of existing works.

**Justification For Why Not Higher Score:**

Comparisons against weak baselines. Lack of clarity in writing, in particular how it contributes methodological on top of the rich line of existing work on constrained decoding.

**Justification For Why Not Lower Score:**

N/A

---

### Decision · Program_Chairs · 2024-01-16

Reject